# Investigation of CO_2_ Adsorption on Avocado Stone-Derived Activated Carbon Obtained through NaOH Treatment

**DOI:** 10.3390/ma16124390

**Published:** 2023-06-14

**Authors:** Joanna Siemak, Rafał J. Wróbel, Jakub Pęksiński, Beata Michalkiewicz

**Affiliations:** 1Department of Catalytic and Sorbent Materials Engineering, Faculty of Chemical Technology and Engineering, West Pomeranian University of Technology in Szczecin, Piastów Ave. 42, 71-065 Szczecin, Poland; joanna.siemak@zut.edu.pl (J.S.); rafal.wrobel@zut.edu.pl (R.J.W.); 2Faculty of Electrical Engineering, West Pomeranian University of Technology, 26 Kwietnia St. 10, 71-126 Szczecin, Poland; jpeksinski@zut.edu.pl

**Keywords:** CO_2_ adsorption, carbon capture, avocado stone, activated carbons, selectivity

## Abstract

Activated carbons were prepared from avocado stone through NaOH activation and subsequent carbonization. The following textural parameters were achieved: specific surface area: 817–1172 m^2^/g, total pore volume: 0.538–0.691 cm^3^/g, micropore volume 0.259–0.375 cm^3^/g. The well-developed microporosity resulted in a good CO_2_ adsorption value of 5.9 mmol/g at a temperature of 0 °C and 1 bar and selectivity over nitrogen for flue gas simulation. The activated carbons were investigated using nitrogen sorption at −196 °C, CO_2_ sorption, X-ray diffraction, and SEM. It was found that the adsorption data were more in line with the Sips model. The isosteric heat of adsorption for the best sorbent was calculated. It was found that the isosteric heat of adsorption changed in the range of 25 to 40 kJ/mol depending on the surface coverage. The novelty of the work is the production of highly microporous activated carbons from avocado stones with high CO_2_ adsorption. Before now, the activation of avocado stones using NaOH had never been described.

## 1. Introduction

Statistically, avocados are experiencing heightened demand, as substantiated by empirical evidence derived from comprehensive analyses of avocado production and harvesting regions. The world’s avocado production reached 1.5 × 10^6^ t in 1980, and in 2021, it reached 8.6 × 10^6^ t [1]. The harvest area is also growing from one year to another. In 1980, the harvest area of avocados was equal to 180,000 hectares, and it was 858,000 hectares in 2021. In the future, a potential increasing demand will be observed. The world’s biggest avocado producer is Mexico (2,442,945 t in 2021). The second and third biggest avocado producers in the world are Colombia (979,618 t in 2021) and Peru (777,096 t in 2021)

Commercial applications exclusively utilize the avocado pulp, disregarding any practical use for other elements of the fruit, such as the stone and peel, resulting in their disposal through landfill. Avocado stones, which account for approximately 26% of the fruit’s total weight, are generated in significant quantities at centralized avocado transformation plants [2]. Despite their substantial starch content, the stones cannot be utilized as livestock feed due to their elevated polyphenol concentration, which imparts a bitter taste and may pose toxicity risks at high levels. In Mexico, 5% of produced avocado was destined for processing (primarily for guacamole), yielding 20,000 t of waste [3]. One way to utilize avocado waste may be the production of an economical and eco-friendly adsorbent material. However, there has been insufficient recognition of its potential as an adsorbent and precursor for the production of activated carbon [3].

Emerging research highlights the successful utilization of avocado waste as a cleaner and more sustainable raw material in the creation of adsorbents for wastewater treatment.

A method for producing activated carbons through activation with H_3_PO_4_ within the temperature range of 800 to 1000 °C was detailed [4]. The resulting material, which exhibited the highest adsorption capacity for blue 41 dye, had a relatively low surface area of 143 m^2^/g and a low pore volume of 0.073 cm^3^/g.

Sulfuric acid was employed as an activating agent in the synthesis of activated carbons from avocado stones, utilizing a temperature of 100 °C [5]. The resulting material exhibited a low specific surface area of 14 m^2^/g and a low pore volume of 0.0323 cm^3^/g and was utilized for the adsorption of Cr(VI) ions from aqueous solutions.

In one study, avocado stones were applied for activated carbon production for phenol removal from water [6]. The physical activation by CO_2_ was performed at a temperature of 900 °C. The obtained activated carbon showed a low specific surface area (206 m^2^/g) and low porosity.

In another study, ZnCl_2_ was used as an activator in a microwave oven [7] for activated carbon production from avocado stones for resorcinol and 3-aminophenol removal from aqueous solutions. The low porosity of the obtained materials was identified (volume of mesopores: 0.325 cm^3^/g and micropores: 0.119 cm^3^/g).

The production of activated carbons from avocado stones through carbonization in nitrogen or carbon dioxide at 600–1000 °C has been presented [8]. The obtained materials exhibited low specific surface area (52–300 m^2^/g) and low pore volume (0.051–0.172 cm^3^/g), especially micropores (0.019–0.122 cm^3^/g). Such properties allowed the use of these activated carbons as sorbents of fluorine ions from aqueous solutions.

All the activated carbons produced from avocado stones exhibited low porosity and were promising sorbent for wastewater. The adsorption of gases over activated carbons produced from avocado stones has not been described before. It was proven that high CO_2_ adsorption requires highly microporous sorbents [9,10,11]. 

Activated carbon production has employed physical methods (thermal, steam) or alkalines (KOH, NaOH), acids (HCl, HNO_3_, H_2_SO_4_, H_3_PO_4_), and other activating agents (O_3_, H_2_O_2_, K_2_CO_3_, ZnCl_2_, FeCl_3_) [12]. KOH [13,14,15,16] was the most often employed, while the others have been less frequently applied. 

NaOH was recently used by research groups as an activation agent. Zhu and Kolar [17] and Cazetta et al. [18] reported that KOH performed very well in terms of the adsorption of p–cresol and methylene blue, respectively. Martins et al. [19] described activated carbon synthesis using NaOH. They demonstrated high surface area and the presence of basic functional groups on the surface. Tan et al. [20] utilized NaOH for the modification of commercial activated carbon to improve their CO_2_ capture. The increase in CO_2_ adsorption was interpreted by the changes in morphology and replacing acid groups with Na^+^.

On the basis of the investigations described above, we hypothesized that using NaOH as an activating agent for avocado stones as a carbon source may be a good solution for efficient CO_2_ sorbents. NaOH has never been used as an activating agent for avocado stones before.

Biomass, with the exception of avocado stones, has been successfully used to produce good CO_2_ sorbents. Table 1 shows the CO_2_ adsorption at a temperature of 0 °C and pressure of 1 bar achieved by other authors.

The objective of this study is to produce activated carbons from avocado stones and NaOH as an activating agent for CO_2_ adsorption. As far as we know, there have been no studies in the literature that demonstrate the use of NaOH as an activating agent for creating activated carbons from avocado stones. The utilization of NaOH resulted in our obtaining activated carbons with high porosity, especially microporosity, which is essential for CO_2_ adsorption. The application of activated carbon from avocado stones for CO_2_ capture is also presented for the first time. A very high CO_2_ adsorption value (qCO_2_0C_) was achieved at a temperature of 0 °C and pressure 1 bar, namely 5.9 mmol/g. Compared to other researchers, this is one of the higher adsorption values.

## 2. Materials and Methods

Activated carbon was prepared from avocado stone. NaOH was applied as activating agent. The temperature of carbonization ranged from 750 to 850 °C. The sodium residues were removed during washing with water. Soaking in 1 M HCl lasted 15 min. The synthesis details are shown in Figure 1.

We collected avocado stones, dried them, and ground them to a powder. On the basis of EDX (Spectrometer XPS, SES-2002, Scienta Scientific AB, Uppsala, Sweden, 2002) of dried powdered material, we can conclude that the precursor was carbonaceous material containing about 5% K (Appendix A). Cr was identified because in order to make these measurements, the sample had to be covered by Cr.

The specific surface area (SSA), total pore volume (V_tot_), micropore volume (V_micro_), pore size distribution, and the highest pore size maxD estimated using the N_2_ adsorption for which the total pore volume was determined were measured on a ASAP Sorption Surface Area and Pore Size Analyzer (ASAP 2460, Micrometrics, Norcross, GA, USA, 2018). Before the measurements, samples were degassed at a temperature of 250 °C for 16 h. Sorption isotherms were investigated at −196 °C. SSAs were calculated based on the BET method. Micropore volume and pore size distribution were calculated through density functional theory using the carbon slit model. Pore volume was estimated on the basis of nitrogen volume adsorbed at the highest p/p_0_.

An X’Pert–PRO, Panalytical, Almelo, The Netherlands, 2012, X-ray diffractometer equipped with a copper source was employed for the XRD analysis. In each XRD scan, the scan rate was 1°/min with 2θ from 5 to 60°.

The field emission scanning electron microscopy images were acquired using a SU8020 Ultra-High Resolution Field Emission Scanning Electron Microscope; Hitachi Ltd., Tokyo, Japan, 2012, under 15 kV voltage.

A sorption analyzer ASAP Sorption Surface Area and Pore Size Analyzer (ASAP 2460, Micrometrics, Norcross, GA, USA, 2018) was also utilized to investigate the nitrogen and carbon dioxide uptake at a temperature of 0 and 20 °C. The adsorption isotherms were obtained.

## 3. Results and Discussion

Nitrogen adsorption–desorption isotherms are presented in Figure 1. The sorption isotherms display I and IV combined types according to the IUPAC [29] classification, meaning that the activated carbons were micro–mesoporous materials.

The significant N_2_ uptake at a relative pressure (p/p_0_) below 0.01 is evident and proved narrow micropore presence. The capillary condensation step at a relative pressure range above 0.55 is present, which demonstrates the presence of mesopores.

According to UPAC, pores are classified into the following groups: micropores < 2.0, mesopores in the range of 2.0–50 nm, and macropores > 50 nm [30]. Some authors subdivide micropores into two groups: narrow micropores or ultramicro pores < 0.7 nm and supermicropores in the range of 0.7 to 2.0 nm [30].

The pore size distribution (Figure 2) of all samples exhibited similar pore structure. The narrow micropores were predominant in all the activated carbons. Textural parameters are listed in Table 2. For all samples, the micropores accounted for about 50% of the pore volume. The highest specific surface area, pore volume, and micropore volume were obtained for C_NaOH_800.

This material also exhibited the highest volume of pores smaller than 1 nm. Activated carbon produced at 800 °C was the most porous material. At lower temperatures, the porous structure was not able to develop well. At higher temperatures, the structure was destroyed to some extent. 

Figure 3 shows SEM images of avocado stone-derived activated carbon obtained through NaOH treatment. Visual observation of the material’s surface revealed the developed surface and macropores that, deeper into the material, may branch into mesopores and finally into micropores.

XRD patterns of avocado stone-derived activated carbon activated by NaOH are shown in Figure 4. Two broad signals at about 22 and 44° are seen. The first peak is attributed to the surface of the turbostratic carbon [31]. The second one is characteristic of the longitudinal dimension, the so-called aromatic sheets [32]. Based on the shapes of the peaks, one can conclude that the avocado stone-derived activated carbons were primarily amorphous [33]. The increase in diffraction intensity at 2θ < 10° confirms the development of micropores and mesopores [34,35]. Apart from broad carbon signals, no other peaks were observed. It can be assumed that sodium was wholly removed from the samples. The EDX results confirmed the absence of Na (Appendix A). 

Figure 5 presents the carbon dioxide adsorption isotherms at 0 °C. The highest CO_2_ adsorption (5.9 mmol/g) was achieved over C_NaOH_800. This material exhibited the highest specific surface area, total pore volume, and micropore volume. The importance of the textural parameters for high CO_2_ adsorption was emphasized by many authors [9,36,37]. In particular, pores with a diameter of less than 1 nm are considered crucial for CO_2_ adsorption [38,39,40,41]. The CO_2_ adsorption values at the temperature of 0 °C and pressure of 1 bar (q_CO2_0C_) are presented in Table 2.

In order to calculate the CO_2_ adsorption selectivity over N_2_ at the temperature of 20 °C, CO_2_ and N_2_ adsorption measurements over C_NaOH_800 were performed at 20 °C. The results of CO_2_ and N_2_ adsorption at 20 °C are presented in Figure 6. The CO_2_ adsorption at the temperature of 20 °C and pressure of 1 bar was equal to 4.3 mmol/g. The decrease in CO_2_ adsorption by increasing the temperature is due to physisorption. To clearly show the effect of CO_2_ adsorption temperature over C_NaOH_800 on CO_2_ adsorption value, Appendix A was drawn.

The CO_2_ and N_2_ adsorption data were fitted by the following adsorption isotherm equations: Langmuir, Freundlich, Sips. These equations were described in [42,43]. The data were more in line with the Sips model. The Sips model is given by the equation:$$q=\frac{q_{m}\cdot b\cdot p^{n}}{1+b\cdot p^{n}}$$
where:*q*—gas equilibrium adsorption at pressure *p*;*p*—equilibrium pressure;*q_m_*—saturation capacity;*b*—equilibrium constant;*n*—exponential parameter representing the heterogeneity of the material.

The fitting accuracy of the models was judged using the least-squares method (LSM), which is most commonly used as an error function [44]: $$LSM=\overset{N}{\underset{i=1}{\sum }}{\left(q_{e,o}-q_{e,z}\right)}^{2}$$
where:*q_e_*_,*o*_—theoretical adsorption capacity calculated from the model;*q_e_*_,*z*_—adsorption capacity determined experimentally;*N*—total number of measurements.

The fitting parameters of the Sips model are displayed in Table 3.

The saturation capacity becomes lower when the temperature becomes higher, which confirms the physical adsorption. The n values are higher than 0.5, which suggests homogeneity of the surface for CO_2_ and N_2_ adsorption. 

Judgment as to the physical or chemical mechanism of adsorption should only be made on the basis of the best fit model. However, the results obtained using the Langmuir model were similar. The value of the saturation capacity of the Langmuir model (q_mL_) for the temperature of 0 °C was also higher than for the temperature of 20 °C for C_NaOH_800. The q_mL_ values confirmed the physical nature of CO_2_ adsorption. The Freundlich model does not describe a limit in adsorption capacity. In this case, the adsorption theoretically may be infinite, so a conclusion similar to the Sips and Langmuir models about the nature of adsorption cannot be drawn.

In order to calculate the selectivity, the Sips model and ideal adsorbed solution theory (IAST) proposed for the first time by Myers and Praunitz [44] were utilized. The IAST was applied to calculate the selectivity of carbon dioxide over nitrogen at 20 °C. The IAST is universally utilized for anticipating the selectivity from mixed gas adsorption isotherms compared to pure component isotherms with a reasonable accuracy for different gas mixtures. The precision of the IAST method was proved for the adsorption of various gas systems over various sorbents [45,46].

The selectivity of gas 1 over gas 2 is possible to predict based on single adsorption isotherms of gas 1 and gas 2:$$S_{\left(g1\right)}=\frac{\frac{x_{g1}}{y_{g1}}}{\frac{x_{g2}}{y_{g2}}}$$
where: *x*_*g*1_ (*x*_*g*2_)—the molar fractions of gas 1/gas 2 (*g*1/*g*2) in the adsorbed phase;*y*_*g*1_, (*y*_*g*2_)—the molar fractions of gas 1/gas 2 (*g*1/*g*2) in the bulk phase. 

The selectivity calculations were made for equimolar mixtures. The results are presented in Figure 7.

The selectivity of CO_2_ ranged from 27 to 6 at pressures from 0.01 to 1. The course of the curve is typical for CO_2_ over N_2_ selectivity [47,48,49]. 

For the CO_2_ captured from flue gas, the selectivity for a mixture containing 15% CO_2_ is essential. The selectivity of CO_2_ over N_2_ for 15% CO_2_ content was calculated based on IAST theory according to the following equation:$$S_{\left({CO}_{2}@15bar\right)}=\frac{q_{{CO}_{2}@15bar}}{q_{N_{2}@85bar}}\cdot \frac{0.85}{0.15}$$

The selectivity for CO_2_ content typical for flue gas was relatively high and was equal to 15, indicating its potential in post-combustion CO_2_ capture.

The isosteric heat of adsorption *Q_iso_* is significant for the evaluation of sorbents. *Q_iso_* gives knowledge about the changes in the enthalpy during the progress of CO_2_ adsorption. It is a measure of the interaction of adsorbate molecules and adsorbent atoms. The isosteric heat of adsorption is expressed by the Clausius–Clapeyron equation:$$Q_{iso}=-R{\left(\frac{\partial ln(p)}{\partial \left(\frac{1}{T}\right)}\right)}_{\theta }$$

*Q_iso_* is the isosteric heat of adsorption at constant surface coverage (kJ/mol), *R* is the gas constant (J/mol·K), and *θ* is the surface coverage degree.

After differentiation of the above equation, a linear equation (adsorption isostere) is obtained:$${\ln\left(p\right)}_{\theta }=-\frac{Q_{iso}}{R}\frac{1}{T}+C$$

Sips equations for temperatures 0 and 20 °C were utilized to calculate the pressure for every value of C_NaOH_800 surface coverage degree. Only two temperatures were applied for the isosteric heat of adsorption calculation, but in [50], it was proven that the results obtained for two and for five temperatures were nearly identical. Adsorption isosteres with different surface coverage are presented in Appendix A.

The linear form was utilized to calculate the isosteric heat of adsorption for various degrees of coverage of the C_NaOH_800 surface with CO_2_ particles. On the basis of the slope (*S*) of the line, the isosteric head of adsorption was calculated for every surface coverage degree.
$$S=-\frac{Q_{iso}}{R}$$

The isosteric heat of adsorption as a function of surface coverage is shown in Figure 8. The *Q_iso_* change is in the range of 25 to 40 kJ/mol, which proves physisorption [51]. The high initial isosteric heat of adsorption is likely caused by stronger Van der Waals’ forces between the carbon surface and CO_2_ molecules at coverage close to 0. With the increase in surface coverage, the heat of adsorption decreases because of the continuous occupation of “CO_2_-philic” active sites [52].

There are two common checkpoints used to determine if adsorption is physical or chemical:-the changes in the gas absorption values (and the saturation capacity) with the increase in the temperature;-the isosteric heat of adsorption value.

If the gas absorption values (and the saturation capacity) decrease with the increase in the temperature, physical adsorption has to be postulated. The heat of adsorption in physisorption lies in the range of 10–40 kJ/mol. If it is higher, the chemical reaction between the gas and surface can be assumed. We showed that the adsorption of CO_2_ values at the same pressure, as well as the saturation capacity, decreased with the increase in the temperature, and the isosteric head of adsorption was lower than 40 kJ/mol. These facts proved that physisorption took place.

## 4. Conclusions

The carbonization temperature of avocado stone-derived activated carbon activated by NaOH substantially influenced the porosity of the resultant materials. For the production of suitable adsorbents for CO_2_, a relatively moderate activation temperature (800 °C) is required. Carbonization at 800 °C provided a high volume of small micropores (size < 1 nm), which is essential for CO_2_ adsorption. 

Activated carbon produced through the NaOH activation of avocado stones and carbonization at 800 °C showed high CO_2_ adsorption (5.9 mmol/g at 0 °C, 1 bar). The Sips isotherm model was found to be the one fitting the adsorption data the best. On the basis of the adsorption, values change with the temperature increase. With the saturation capacity of the Sips equation and the isosteric heat of adsorption, the physisorption of CO_2_ was found over avocado stone-derived activated carbon activated by NaOH.

A very high CO_2_ adsorption value at 0 °C and pressure of 1 bar was achieved: 5.9 mmol/g. Compared to other researchers, this is one of the higher adsorption values.

## Data Availability

The data presented in this study are available upon request from the corresponding author.

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
