# Peer review of "Investigation of CO2 Adsorption on Avocado Stone-Derived Activated Carbon Obtained through NaOH Treatment"

_materials, 2023, doi:10.3390/ma16124390_

Round 1

Reviewer 1 Report

 The work on CO2 absorbing material preparation by  Siemak et al is not done in a proper way. The manuscript at this stage can not be recommended for publication. I suggest working on the following problems before consideration.

 â€‹1. The incorrect usage of the degree symbol is observed in lines 47 and 119.

​2.  The figure 1 should be represented as Scheme 1.

3. After the number, there should be a space to write units.

4. At the results and discussion part, the template instructions are probably still there on lines 67-69.

5. The XRPD pattern of any amorphous compound can not be explained by interpreting them with the lattice plane. If there is no lattice cell, how can the plane arise? Until you solve a crystal structure, you can never assign any peak with some lattice plane. It is preferable to interpret them by the distances between the layers. 

6. The sample without NaOH should be compared with the NaOH-treated sample to explain the contribution of NaOH.

7. Is there any probability of the sodium ion to stay intercalated between the two layers? EDX for elementary mapping is essential.

8. Does Na+ play any pivotal role to bind CO2 here? Thus, elemental analysis is very important.

9. The explanation of the CO2-absorbing mechanism is not very convincing.

10. SEM image at micrometre level is not useful to study the high-resolution morphology.

11. The introduction part is not written well. 

12. Where is the unit of pressure at the line 76 and 78?

The quality is not bad. 

Author Response

Response to Reviewer 1 Comments

The work on CO2 absorbing material preparation by  Siemak et al is not done in a proper way. The manuscript at this stage can not be recommended for publication. I suggest working on the following problems before consideration.

  1. The incorrect usage of the degree symbol is observed in lines 47 and 119.

Response:

It was changed.

  1. The figure 1 should be represented as Scheme 1.

Response:

It was changed.

  1. After the number, there should be a space to write units.

Response:

It was changed.

  1. At the results and discussion part, the template instructions are probably still there on lines 67-69.

Response:

It was removed.

  1. The XRPD pattern of any amorphous compound can not be explained by interpreting them with the lattice plane. If there is no lattice cell, how can the plane arise? Until you solve a crystal structure, you can never assign any peak with some lattice plane. It is preferable to interpret them by the distances between the layers.

Response:

Yes, you are right. We remove this information about the lattice plane.

  1. The sample without NaOH should be compared with the NaOH-treated sample to explain the contribution of NaOH.

Response:

The samples NaOH treated didn’t contain any NaOH or Na compound. All the Na ions and compounds were washed. You can see it in Figure S 1. Sodium is not present at the EDX spectra. The role of NaOH is to react with the carbonaceous materials and produce microporous structure.

After carbonization of avocado stone without NaOH treating, non-porous char was obtained.

We tried to measure the textural parameters of this material but it was not possible to obtain the correct isotherm. The value of the specific surface area was below the capabilities of the apparatus, that’s means it was lower than 1 m2/g. The CO2 adsorption measures also failed.

  1. Is there any probability of the sodium ion to stay intercalated between the two layers? EDX for elementary mapping is essential.

Response:

As I explained in point 6, there is no sodium in the activated carbons. The EDX spectra were added to the manuscript.

  1. Does Na+ play any pivotal role to bind CO2 here? Thus, elemental analysis is very important.

Response:

As I explained in point 6, there is no sodium in the activated carbons. The EDX spectra were added to the manuscript.

  1. The explanation of the CO2-absorbing mechanism is not very convincing.

Response:

The CO2 adsorption is definitely physical.

There are two common checkpoints if adsorption is physical or chemical:

- the changes of the gas absorption  values (and the saturation capacity) with the increase of the temperature

- the isosteric heat of adsorption value

If the gas absorption value (and the saturation capacity) decreases with the increase of the temperature, physical adsorption has to be postulated. The heat of adsorption in physisorption lies in the range of 10−40 kJ/mol. If it is higher, the chemical reaction between gas and surface can be assumed.

            We showed that the adsorption of CO2 value, as well as the saturation capacity, decreases with the increase of the temperature and the isosteric head if adsorption was lower than 40 kJ/mol. These facts proved that physisorption took place.

  1. SEM image at micrometre level is not useful to study the high-resolution morphology.

Response:

Yes, you are right, but the readers can note some macropores and that the morphology of materials was quite similar. Of course, if you insist, we can remove it. Unfortunately, we are not able to make TEM micrographs.

  1. The introduction part is not written well.

Response:

The introduction was completely changed.

  1. Where is the unit of pressure at the line 76 and 78?

Response:

There is no unit because it is relative pressure (p/p0). The p/p0 was added in brackets for clarity.

Reviewer 2 Report

In this manuscript, the results of this research are conveyed thoughtfully and completely, and they are consistent with the experimental findings. However, the authors failed to explain and draw out the novelty of the work, this aspect needs to be improved. This work is worthwhile to be publish in this journal after major revision. The following issues should be addressed:

1. Introduction is very short, also, the importance and novelty of the research should be highlighted and more clearly stated. The authors should give some examples of works in the bibliography, to clear the advantage of their work in comparison with those works.

2. Maybe the author should compare their results clearly with other reported works, highlighting the advantage and disadvantages of their novel composite.

3. Authors should look for language errors carefully. They have been spotted throughout the manuscript.

4. Introduction part, some important and relative references could help to define prepare the activated carbon from biomass wastes and also using different materials to investigate its application to remove different pollutants

https://doi.org/10.1007/s10904-023-02604-0, https://doi.org/10.1016/j.est.2023.107168, https://doi.org/10.1016/j.est.2023.106806

5. The abstract written in two paragraphs, why? And also, it need some paraphrasing to show the novelty.

6. In Materials and method section, please provide the purity of your chosen precursors.

Hence, I recommend it accepted for publication after Major revisions.

Author Response

Response to Reviewer 2 Comments

In this manuscript, the results of this research are conveyed thoughtfully and completely, and they are consistent with the experimental findings. However, the authors failed to explain and draw out the novelty of the work, this aspect needs to be improved. This work is worthwhile to be publish in this journal after major revision. The following issues should be addressed:

  1. Introduction is very short, also, the importance and novelty of the research should be highlighted and more clearly stated. The authors should give some examples of works in the bibliography, to clear the advantage of their work in comparison with those works.

Response:

The introduction was completely changed.

  1. Maybe the author should compare their results clearly with other reported works, highlighting the advantage and disadvantages of their novel composite.

Response:

Table 1. CO2 adsorption at a temperature of 0 °C and pressure of 1 bar obtained with the other authors was added to the introduction in order to compare with the activated carbon produced from avocado seeds. 

  1. Authors should look for language errors carefully. They have been spotted throughout the manuscript.

Response:

The quality of English in the manuscript was enhanced.

  1. Introduction part, some important and relative references could help to define prepare the activated carbon from biomass wastes and also using different materials to investigate its application to remove different pollutants

https://doi.org/10.1007/s10904-023-02604-0

https://doi.org/10.1016/j.est.2023.107168

https://doi.org/10.1016/j.est.2023.106806

Response:

The manuscripts listed by the reviewer are so far from activated carbon from biomass wastes and even removing pollutants. Because of it, I’m not going to cite these manuscripts.

https://doi.org/10.1007/s10904-023-02604-0

Tayseer M. Alasri, · Shaimaa L. Ali, · Reda S. Salama, · Fares T. Alshorifi

Band‑Structure Engineering of TiO2 Photocatalyst by AuSe Quantum Dots for Efficient Degradation of Malachite Green and Phenol

It is not about activated carbon. It is not about removing pollutants but degradation

https://doi.org/10.1016/j.est.2023.107168

Mohamed Mokhtar M. Mostafa, Abdelmohsen A. Alshehri , Reda S. Salama

High performance of supercapacitor based on alumina nanoparticles derived from Coca-Cola cans

It is not about activated carbon produced from biomass waste. It is not about removing pollutants.

https://doi.org/10.1016/j.est.2023.106806

Mostafa S. Gouda, Mona Sheha, Shacker Helmy, Moataz Soliman, Reda S. Salama

Nickel and cobalt oxides supported on activated carbon derived from willow catkin for efficient supercapacitor electrode

It is about activated carbon produced from biomass, but nickel and cobalt oxides were supported on it. It is not about removing pollutants.

  1. The abstract written in two paragraphs, why? And also, it need some paraphrasing to show the novelty.

Response:

The abstract was changed. Information about novelty was added.

  1. In Materials and method section, please provide the purity of your chosen precursors.

Response:

The precursor was biomass, namely avocado stone. We collected stones, dried them, and ground them to a powder. On the basis of EDX of dried powdered material, we can conclude that the precursor was carbonaceous material containing about 5% K (Figure S1). Cr was identified because in order to make these measurements sample had to be cover by Cr.

This information was added.

Hence, I recommend it accepted for publication after Major revisions.

Reviewer 3 Report

This work proposes the preparation of activated carbon from avocado stones following optimization of the NaOH treatment. Application for selective CO2 adsorption is also presented. The present study is well done and only few minor problems can be noted. I therefore recommend accepting this work after minor revision.

Main comments

1) The title should be improved: the word activated appears twice. Since activated carbon is commonly used to describe this kind of carbon material, I suggest changing the end by: “…activated carbon obtained by NaOH treatment”

2) All along the text and in tables: please keep in mind that values must be given with a realistic degree of precision. Therefore, pore volumes and microporous volumes must be given with only two decimals. Indeed, in Table 2, qm and error values must be given with one decimal.

Minor comments

1) Some typo errors:
- Line 13: results, value of
- Line 14: suppress “(15)”
- Line 27: and other activating agents
- Line 36: adsorption was interpreted by changes, Na+
- Line 47: 850°C
- Line 74: types
- Line 75: meaning that
- Line 96: “numerous of macropores”? English to be revised
- Line 102: 44°
- Line 112: presents the
- Line 119: 20°C
- Line 151: for anticipating the selectivity from mixed gas adsorption isotherms compared to pure component
- Line 196: was found to be the one fitting the best the adsorption data
- Line 197: of the Sips equation and the isosteric heat of adsorption, physisorption of CO2 was found

2) Figure 1: indicate the duration for the HCl soaking step

3) The text in lines 67-69 must be suppressed.

4) Line 94: same remark as for the title

5) Line 123: is due to physisorption

Only few errors in English to be noted, all of them are listed in the review

Author Response

Reviewer 3

This work proposes the preparation of activated carbon from avocado stones following optimization of the NaOH treatment. Application for selective CO2 adsorption is also presented. The present study is well done and only few minor problems can be noted. I therefore recommend accepting this work after minor revision.

Main comments

1) The title should be improved: the word activated appears twice. Since activated carbon is commonly used to describe this kind of carbon material, I suggest changing the end by: “…activated carbon obtained by NaOH treatment”

Response:

You are right. It was changed

2) All along the text and in tables: please keep in mind that values must be given with a realistic degree of precision. Therefore, pore volumes and microporous volumes must be given with only two decimals. Indeed, in Table 2, qm and error values must be given with one decimal.

Response:

I cannot agree. All authors give pore volumes and microporous volumes with three decimals. It is very common. I cannot find manuscript with only two decimals.

The qm precision and error values were come from the statistic. Errors were ranged from 0,004 to 0.00003. I give them with one decimal.  However 4.33∙10-03 was changed for 4∙10-03 etc.

Minor comments

1) Some typo errors:
- Line 13: results, value of
- Line 14: suppress “(15)”
- Line 27: and other activating agents
- Line 36: adsorption was interpreted by changes, Na+
- Line 47: 850°C
- Line 74: types
- Line 75: meaning that
- Line 96: “numerous of macropores”? English to be revised
- Line 102: 44°
- Line 112: presents the
- Line 119: 20°C
- Line 151: for anticipating the selectivity from mixed gas adsorption isotherms compared to pure component
- Line 196: was found to be the one fitting the best the adsorption data
- Line 197: of the Sips equation and the isosteric heat of adsorption, physisorption of CO2 was found

Response:

We made all the suggested changes

2) Figure 1: indicate the duration for the HCl soaking step

Response:

Information was added above the Fig. 1.

3) The text in lines 67-69 must be suppressed.

Response:

It was changed

4) Line 94: same remark as for the title

Response:

It was changed

5) Line 123: is due to physisorption

Response:

It was changed

Comments on the Quality of English Language

Only few errors in English to be noted, all of them are listed in the review

Reviewer 4 Report

This work informs on the preparation of a new carbon dioxide sorbent and its basic properties. Methods are standard, the raw material is perhaps new as well as the activating agent, though these novelties are trivial. Nevertheless, the produced material and its sorption properties may be interesting for those involved in carbon dioxide capture. I have following comments:

·         It would be nice if authors inform readers (in introduction) why just avocado stone, what are its advantages, availability and so on. Further, the essence of the activation role of NaOH should be mentioned, then, why sorbents with and without NaOH were not compared.

·         Lines 67-69 are superfluous.

·         I recommend adding more details on BET and DFT calculations, preferably in supplementary information for readers to be able to see and check the calculation procedures.

·         The dimensions of various pore types (size limits), micro-/macro-, should be given explicitly regardless the common use of this classification.

·         Please, explain (previously not explained) symbols used in table 1.

·         Sorption properties are crucial for this work. I thus highly recommend to add (supplementary information) more information on the sorption isotherm measurements and construction (e.g., figure 6), including information pressure of what exactly is shown in figures and the calculations of the isosteric heat of adsorption (are two temperatures sufficient?). Among others, for potential reproducibility measurements of interested readers.

·         What exactly is meant by “Error” in table 2?

·         The effect of temperature on adsorption is seen in table 2 only and only on parameters of the Sips model. What was the temperature effect on the other measured characteristics combined with the effect of the carbonization temperature?

·         In conclusions I would expect a comparison with some other sorbents and their performance.

Author Response

Reviewer 4

This work informs on the preparation of a new carbon dioxide sorbent and its basic properties. Methods are standard, the raw material is perhaps new as well as the activating agent, though these novelties are trivial. Nevertheless, the produced material and its sorption properties may be interesting for those involved in carbon dioxide capture. I have following comments:

  • It would be nice if authors inform readers (in introduction) why just avocado stone, what are its advantages, availability and so on. Further, the essence of the activation role of NaOH should be mentioned, then, why sorbents with and without NaOH were not compared.

Response:

The introduction was changed entirely. Information about avocado advantages, availability, and so on were also added.

We didn’t use sorbents with NaOH. NaOH was an activating agent and, during washing, was removed entirely. Have a look at Figure S1. There is no Na signal in the EDX spectra.

  • Lines 67-69 are superfluous.

Response:

It was removed.

  • I recommend adding more details on BET and DFT calculations, preferably in supplementary information for readers to be able to see and check the calculation procedures.

Response:

The specific surface area calculation

Specific surface area (SSA) was calculated on the basis of the BET equation in the range of partial pressure of p/p0=0.05-0.2. This range was narrowed individually for each material so that a linearity of BET plot will be fulfilled. The function:

was plotted. W is the mass of nitrogen adsorbed at a relative pressure p/p0, Wm is the mass of N2 constituting a monolayer, p is the partial pressure of nitrogen, p0 is the saturated vapour pressure of nitrogen under the temperature of 77 K (N2 dew point). For microporous materials, the linear region is shifted to lower relative pressures.

Wm was calculated on the basis of slope (s) and the intercept (i) of the linear region of the function.

Wm was calculated by combining the above equations:

The specific surface area was calculated by:

N is Avogadro’s number (6.023∙1023 molecules/mol), M is the molecular weight of N2 (28.02 g/mol), A is the molecular cross-sectional area N2 molecule (1.62 nm2).

The range of linear region was identified by authors, and the SSA was calculated by the software ASAP 2460 version 3.01.

DFT method

The DFT models are created by classical approaches to adsorption as well as models based on modern statistical thermodynamics. There are 14 DFT models in the software ASAP 2460 version 3.01, depending on the pore shape and adsorbent type. The authors chose the best model on the basis of adsorbent properties and goodness of fit vs. regularization. All the calculations were performed in the software ASAP 2460 version 3.01.

These information were added to Supplementary materials

  • The dimensions of various pore types (size limits), micro-/macro-, should be given explicitly regardless the common use of this classification.

Response:

This information was added:

According to UPAC pores are classified into the following groups: micropores < 2.0, mesopores in the range of 2.0 - 50 nm, macropores > 50 nm. Some authors subdivide micropores into two group: narrow micropores or ultramicropores <0.7 nm and supermicropores in the range of 0.7 to 2.0 nm.

  • Please, explain (previously not explained) symbols used in table 1.

Response:

The symbols were explained in Materials and Methods section

  • Sorption properties are crucial for this work. I thus highly recommend to add (supplementary information) more information on the sorption isotherm measurements and construction (e.g., figure 6), including information pressure of what exactly is shown in figures and the calculations of the isosteric heat of adsorption (are two temperatures sufficient?). Among others, for potential reproducibility measurements of interested readers.

Response:

The CO2 adsorption was performed using ASAP Sorption Surface Area and Pore Size Analyzer (ASAP 2460, Micrometrics, Novcross, USA) 2018. Authors introduced the values of the pressure and the device establish the equilibrium point by point. We checked the reproducibility measurements several times and it was excellent every time. So we don’t do it for each sample.

Isosteric heat of adsorption Qiso is significant for the evaluation of sorbents. Qiso  gives knowledge about the changes in the enthalpy during the progress of CO2 adsorption. It is a measure of the interaction of adsorbate molecules and adsorbent atoms. The isosteric heat of adsorption is expressed by the Clausius-Clapeyron equation:

Qiso is isosteric heat of adsorption at constant surface coverage [kJ/mol], R is gas constant [J/mol·K], q is surface coverage degree.

After differentiation of the above equation, a linear equation (adsorption isostere) is obtained:

Sips equations for temperatures 0 and 20 °C was utilized to calculate the pressure for every value of C_NaOH_800 surface coverage degree. Only two temperatures were applied for the isosteric heat of adsorption calculation, but in [Serafin, J.; Baca, M.; Biegun, M.; Mijowska, E.; KaleÅ„czuk, R.J.; SreÅ„scek-Nazzal, J.; Michalkiewicz, B. Direct Conversion of Biomass to Nanoporous Activated Biocarbons for High CO2 Adsorption and Supercapacitor Applications. Appl. Surf. Sci. 2019, 497, 143722, doi:10.1016/j.apsusc.2019.143722.} was proved that the results obtained for two and for five temperatures were were nearly identical. Adsorption isosteres with different surface coverage were presented in Figure S5.

The linear form was utilized to calculate isosteric heat of adsorption for various degrees of coverage of the C_NaOH_800 surface with CO2 particles. On the basis of the slope (S) of the line, isosteric head of adsorption was calculated for every surface coverage degree.

  • What exactly is meant by “Error” in table 2?

Response:

The meaning of Error is LSM.

To evaluate the best fittings of isotherm models to the experimental data has been applied the least-squares method (LSM), which is most commonly used as error functions [J.C.Y. Ng, W.H. Cheung, G. McKay, Equilibrium studies of the sorption of Cu(II) ions onto chitosan, J. Colloid Interf. Sci., 255 (2002) 64-74. https://doi.org/10.1006/jcis.2002.8664]:

where:

qe,o – theoretical adsorption capacity calculated from the model
qe,z – adsorption capacity determined experimentally

N – total number of measurements

The “Error” was changed for “LSM”.

  • The effect of temperature on adsorption is seen in table 2 only and only on parameters of the Sips model. What was the temperature effect on the other measured characteristics combined with the effect of the carbonization temperature?

Response:

We don’t need the model of the isotherm to judge the effect of temperature adsorption. The conclusion on the temperature effect can be drawn on the basis of the Fig. 5 and 6. The CO2 adsorption at 20 oC is lower than at 0 oC what indicates physical adsorption. In order to show it clearly the adsorption isotherms at 0 and 20 oC for C_NaOH_800 were presented in supplementary materials in the Fig. S2. Unfortunately, we have measured the CO2 adsorption at temperature 20 only for C_NaOH_800 because the CO2 adsorption over this material at 0oC was the best.

We added also the tables containing the Langmuir and Freundlich parameters for C_NaOH_800 and the figures with these models. Langmuir and Freundlich model didn’t fit well.

Judgment as to the physical or chemical mechanism of adsorption should only be made on the basis of the best fit model. But the results obtained using the Langmuir model were similar. The values of saturation capacity of the Langmuir model (qmL) for the temperature of 0 oC were also higher than for a temperature of 20 oC. for C_NaOH_800. The qmL values confirmed the physical nature of CO2 adsorption. Freundlich model does not describe a limit in adsorption capacity. In this case the adsorption theoretically may be infinite so a conclusion similar to Sips and Langmuir models about the nature of adsorption cannot be drawn.

Similar information were added to the manuscript.

  • In conclusions I would expect a comparison with some other sorbents and their performance.

Response:

Table 1. CO2 adsorption at a temperature of 0 °C and pressure of 1 bar obtained with the other authors was added to the introduction in order to compare with the activated carbon produced from avocado seeds. In conclusion, a comparison was added.

Round 2

Reviewer 1 Report

The work is thoroughly modified and addresses all the issues properly. However, the literature on CO2 absorption is remain partial and the review works are relatively old.  I suggest to include the followin papers 

J. Am. Chem. Soc. 2013, 135, 48, 18040–18043

ACS Appl. Mater. Interfaces 2014, 6, 20, 18352–18359

and some recent reviews like 

https://doi.org/10.1007/s13762-022-04680-0.

The cited reviews in the manuscript are very old to cite. So, cite some new reviews. 

Reviewer 2 Report

Accepted in the present form